# OpenReview forum: "PluriHarms: Benchmarking the Full Spectrum of Human Judgments on AI Harm"
_ICLR.cc/2026/Conference — ICLR 2026 Poster_

### Official Review · Reviewer_3ftX · 2025-10-30

**Soundness:** 4
**Presentation:** 4
**Contribution:** 3
**Rating:** 6
**Confidence:** 3

**Summary:**

This paper introduces PLURIHARMS, to move beyond binary safety labels, a safety benchmark aimed at modeling not only harmfulness but also human disagreement about harm. The dataset contains 150 prompts spanning an ordinal harm spectrum, each annotated by 100 crowdworkers (15,000 total ratings) with rich demographic and psychological traits, and coupled with prompt-level features derived from SafetyAnalyst (actions/effects) and KALEIDO (values/rights/duties). Methodologically, the authors curate prompts via LLM-based synthesis and a genetic algorithm to over-sample borderline cases; they analyze determinants of judgments using mixed-effects regressions with lasso selection, and evaluate several safety models and personalization strategies (e.g., value-profile and k-shot steering). Results show (i) human ratings correlate with synthetic harm levels and exhibit higher variance/entropy in the mid-range, (ii) both prompt features and annotator traits (and their interactions) significantly shape harm judgments, and (iii) personalized alignment improves prediction over consensus-based approaches, though headroom remains.

**Strengths:**

- The paper squarely advances the field beyond binary safety labels by treating harm as an ordinal and distributional phenomenon, making disagreement itself a first-class signal rather than noise and motivating pluralistic safety evaluation.
- The authors implement a multi-step data-hygiene pipeline (e.g., attention checks, time-on-task floors, duplicate and consistency screening, and clear exclusion criteria), which materially improves label reliability and reduces typical crowd-sourcing artifacts. And annotator metadata are carefully documented, with clear distributions over demographics and psychological traits; this transparency enables reproducible research on personalization, fairness, and cross-group heterogeneity instead of relying on unverifiable assumptions.
- By showing that annotator traits and their interactions with prompt features explain systematic disagreement, the paper offers a convincing, data-driven account of value pluralism in harm judgments—an insight that is both methodologically careful and theoretically meaningful for personalized alignment.
- The benchmarking of safety models and personalized alignment methods on PLURIHARMS demonstrates consistent gains from personalization over consensus predictors, signaling practical utility today while clearly delineating room for future improvement in sample efficiency and generalization.
- Figures and narrative are clear and well-scoped, with limitations explicitly acknowledged, which increases trust in the findings.

**Weaknesses:**

- The paper commendably reports rich annotator trait distributions (demographic and psychological), but the pool is entirely U.S.-based (as the authors referred in Appendix D), so cultural/linguistic coverage is narrow and may limit the external validity of personalization/pluralism findings; a multilingual, multi-region replication with stratified sampling would strengthen claims.
- The paper posits that "harm" and "values" are the two key dimensions for human harm perception (lines 145-147). While this operationalization is reasonable, the paper does not show that a two-dimensional schema is sufficient or unique; a comparison with alternative ontologies or a data-driven dimensionality analysis would make the claim more compelling.
- Prompt spectrum relies on a single generator (DeepSeek-V3-0324) and AIR-Bench seeds. Although Appendix A specifies an 11-level grid and human ratings show a monotonic correlation with the synthetic levels, the benchmark's mid-spectrum may still be wording-sensitive or generator-specific. Robustness checks would strengthen the claim that disagreement is value-driven rather than phrasing-driven.
- The evaluation probes only three models, which limits conclusions about cross-model generality. For a benchmark that aims to study pluralism and disagreement, broader coverage is important—not only more backbones but also diversity along key axes.

**Questions:**

- During prompt generation (Appendix A), did high-harm prompts (like ≥0.8) frequently trigger safety refusals by the generator? If so, could you share refusal/abort rates by harm level and any backoff strategies (e.g., re-prompting, softer wording)? It would help to know whether refusals biased the realized spectrum (e.g., under-sampling at the top end).
- In Final Dataset Curation (around lines 157–161), you describe targeting the desired harm-level distribution. Could you also report the dataset composition for other key features? This would clarify coverage and potential imbalances beyond the harm level.
- Would you report a small cold-start slice? For example, leave-one-annotator-group-out (by trait strata) or leave-one-topic-cluster-out. This would test whether personalization via traits still helps for unseen users and content.

If the authors can substantively address the questions and weaknesses outlined above, I would be inclined to raise my score.

---

> ### Author Response · Authors · 2025-11-24
> **Response to Reviewer 3ftX**
>
> Thank you again for the thoughtful and constructive feedback. We hope the clarifications and new analyses below address your concerns, and we would be grateful if you would consider raising your score.
>
> # Weaknesses
>
> ### **Weakness 1**
> We agree that our annotator pool, though intentionally diverse along demographic and psychological dimensions, was limited to U.S.-based participants. As noted in the *Limitations* and Appendix D, this restricts cultural and linguistic breadth. Our present focus is to establish a controlled framework that can systematically surface disagreement-prone harm judgments. We encourage future extensions to multilingual and multi-region populations.
>
> ### **Weakness 2**
> We appreciate the request for deeper justification of our choice of harms and values as the main feature dimensions. To address this, we performed an additional PCA-based dimensionality analysis over Qwen3-4B activations (Appendix D.5), providing a high-capacity baseline for comparison against our interpretable SafetyAnalyst and KALEIDO space.
>
> - **PCA with matched dimensionality (42 PCs):** R² = **0.339** vs. **0.273** for our interpretable model.
> - **High-capacity PCA ceiling (119 PCs → 70 after lasso):** R² = **0.391**.
>
> Although high-dimensional embeddings capture more variance, the gains are small relative to dimensionality cost and loss of interpretability. The competitive performance of our structured harm–value space supports its substantive relevance.
>
> ### **Weakness 3**
> To test whether the synthetic harm spectrum might reflect generator-specific wording cues, we conducted a LIWC-based stylistic analysis (see our response to reviewer wR6w for detail). Across 15 stylistic categories, **none showed significant correlations** with human harmfulness ratings after Holm–Bonferroni correction. Even the largest raw correlations were modest. This suggests DeepSeek’s phrasing does not introduce systematic stylistic artifacts.
>
> Additionally, including the synthetic harm level as a regressor in our linear models does not change the qualitative conclusions—any systematic bias tied to specific harm levels would be captured by this term. We omitted this analysis from the paper to avoid redundancy, but would be happy to add it to an appendix if the reviewer feels it would strengthen the manuscript.
>
> ### **Weakness 4**
> We agree broader model coverage is important. In response, we expanded our k-shot evaluation to include **GPT-5**, multiple **Claude** variants (3 → 4.5), and **Qwen3** models from **4B–32B**. Table 1 in the revised manuscript summarizes results.
>
> Key patterns:
> - **Smaller models frequently failed** to output even a short numeric prediction within a 1024-token limit, revealing practical limitations when general LLMs are used for fine-grained harm rating without specialized training.
> - **Some Claude models** occasionally refused harmless numeric predictions, likely due to safety filtering triggered by harmful prompts, despite not asked to produce harmful content.
> - **GPT-5 performed strongest overall**, but the *relative* benefit of personalized over aggregated alignment was stable across all model families.
>
> ---
>
> # Questions
>
> ### **Question 1**
> We observed **no refusals during prompt generation** with DeepSeek-V3, even for high-harm prompts. Refusals occurred only during **evaluation** for some Claude variants (see Table 1), and therefore did not affect dataset creation.
>
> ### **Question 2**
> Thank you for the suggestion. We added **Figure 15 in Appendix D**, which reports the distributions of *harm levels*, *harmful action categories*, *harmful effects*, and *value categories* across the curated prompts. These visualizations clarify both the targeted over-representation of intermediate harm levels and the realized topical distribution. Because our seed pool is drawn from AIR-Bench prompts, some topical biases naturally persist despite our efforts to diversify action–effect–value coverage.
>
> ### **Question 3**
> We agree that cold-start evaluation is an important question for assessing generalization to unseen users or content. However, PluriHarms was designed to maximize **within-annotator** signal rather than across-annotator stratification. With **150 annotators**, we prioritized collecting *dense* per-annotator ratings (all 150 prompts) over expanding the cohort size. As Figure 10 shows, the demographic strata were not intentionally balanced, leaving some groups too small for reliable leave-one-group-out or intersectional slicing. Because the dataset is curated around prompt-level feature diversity rather than annotator-group balance, we currently lack sufficient power for robust cold-start evaluation without risking overinterpretation. We agree it is a valuable direction and view such analyses as a natural extension for future work or larger-scale replications.

---

### Official Review · Reviewer_96EC · 2025-10-31

**Soundness:** 3
**Presentation:** 4
**Contribution:** 4
**Rating:** 8
**Confidence:** 4

**Summary:**

This paper introduces a PluriHarms, a benchmark for safety that consists of 150 prompts with annotations from 100 human annotators from various backgrounds, for a total of 15,000 ratings. PluriHarms aims to capture prompts that have subjective or ambiguous safety ratings, and capture differing views about safety from a diverse range of human annotators.

**Strengths:**

- Annotator demographics are included, and are relatively diverse across a number of demographic categories reported in Appendix D.3.
- The paper argues that disagreement should be treated as diverse, legitimate viewpoints, which is an important principle for pluralism.
- As a benchmark, PluriHarms is challenging; WildGuard only performs about as well as the random baseline. This demonstrates that there is still a gap for developing safety models that can be aligned to diverse viewpoints.
- This paper contributes a concrete dataset that can be used to ground the community’s discussion on pluralism and safety.

**Weaknesses:**

- PluriHarms has only 150 prompts, which is a small number compared to other datasets such as AIR-Bench 2024 with 5,692 prompts.
- More information could have been provided in Section 2 about the genetic algorithm used to curate prompts.
- Annotator recruitment was done on Prolific, thus the annotator sample is subject to the biases of the annotator population on that platform, and may not generalize to other populations.

**Questions:**

- For the final curation of 150 prompts, why was a genetic algorithm chosen over alternative simpler sampling methods?
- Were all annotators based in the United States? If so, this should be acknowledged as a limitation.
- Could you share your assessment of privacy risks associated with releasing annotation and demographic data of the annotators, and how you have mitigated these risks? Was this adequately addressed in your IRB research plan?
- In Figure 8, does each dot represent the MAE of the model against a single annotator?

---

> ### Author Response · Authors · 2025-11-22
> **Response to Reviewer 96EC**
>
> We thank Reviewer 96EC for the thoughtful and constructive feedback, as well as the clear and supportive assessment of our work. We address each weakness and question below.
>
> ## **Weaknesses**
>
> ### **Weakness 1**
> We agree that PluriHarms contains fewer prompts than large-scale datasets like AIR-Bench 2024. However, AIR-Bench is a **synthetic** prompt dataset, whereas PluriHarms is designed specifically around **human annotations** with rich within-annotator information. Our goal is not to maximize breadth of synthetic prompts, but to maximize **depth of annotation per participant**, which is essential for studying personalized and pluralistic alignment. Each participant rated all 150 prompts—a dense within-annotator design that supports individualized modeling in ways synthetic prompt-only datasets cannot. Participants spent an average of 65 minutes on the task, which is comparable to the PRISM alignment dataset.
>
> ### **Weakness 2**
> We appreciate the request for additional detail. As noted in Section 2, we use a *standard* genetic algorithm with equally weighted multi-objective optimization; we did not describe it extensively because it is a well-established method rather than a contribution of our own. It was chosen because our goal required matching **multiple feature distributions simultaneously**, a setting where genetic algorithms are a natural and well-documented choice. We now add a brief clarification to Section 2 of the paper noting that our implementation includes **custom constraint-preserving crossover and mutation operators** (e.g., enforcing a maximum of two prompts per seed set), which ensure that curated prompts satisfy the structural constraints of our dataset. We also clarify that we use the inverse Jensen–Shannon distance between each candidate prompt’s empirical feature distributions and the corresponding targets to compute loss. Finally, upon publication we will release the full code, including the exact genetic algorithm configuration, to facilitate full transparency and reproducibility.
>
> ### **Weakness 3**
> We agree. Since annotators were recruited on Prolific, the sample naturally reflects the demographics and biases of that platform’s participant pool. This limitation applies to all studies conducted on Prolific, and we already acknowledge the constrained demographic coverage in the final paragraph of the main text. We highlight this explicitly and note that future work should extend the framework to more diverse populations to assess generalizability.
>
> ---
>
> ## **Questions**
>
> ### **Question 1**
> We chose a genetic algorithm because our goal was to align the final set of 150 prompts with **multiple simultaneous feature-distribution objectives** (over harm levels, action types, effects, and values). A genetic algorithm is a natural fit for this kind of multi-objective optimization, where simple or heuristic sampling would struggle to satisfy all constraints jointly. Since this curation step was ran offline, computational cost was not a concern, allowing us to prioritize **quality of fit** over simplicity in choosing the algorithm. As a result, the genetic algorithm provided a reliable way to obtain a diverse prompt set focused on borderline cases that met our specified targets.
>
> ### **Question 2**
> Yes. We now acknowledge this in the limitations section more explicity: all annotators were U.S.-based, which constrains cultural diversity in the dataset. We encourage future work to apply this framework cross-culturally to capture broader variation in harm perceptions.
>
> ### **Question 3: Privacy Risks**
> Thank you for encouraging us to clarify the privacy precautions we’ve taken to minimize risks to our human participants. This study was approved by our IRB (IRB number redacted for blind review). All data will be anonymized before release and the data collected does not uniquely identify individuals. Although we collected non-standard demographic data (e.g., religion, political leaning, online toxicity experience), those are not more identifying than standard demographic data (e.g., race, gender, age). It is possible that releasing individual annotator ratings on psychological questionnaires might increase risk should annotators somehow be identified, so we will only release them in composite factors (the 3 factors derived from our factor analysis and used in all our analyses) rather than individual scores.
>
> ### **Question 4**
> Yes. Each dot in Figure 8 represents the MAE of the model evaluated against a single annotator’s ratings — and we show individual-level results to make clear that alignment methods do **not** work equally well for everyone.

---

> > ### Comment · Reviewer_96EC · 2025-11-24
> >
> > - W1: Thanks for the explanation. I agree that the number of instances is constrained by the requirement of having full annotator coverage of all instances.
> > - W2 & Q1: Thank you for providing more information regarding the design and justification of the genetic algorithm. I am not very confident of my assessment of the algorithm due to my lack of expertise in this area, but I find your explanation to be satisfactory, and I appreciate the inclusion of additional details and code for this method.
> > - W3 & Q2: Thank you for acknowledging the limitation regarding US-based annotators.
> > - Q3: Thank you on sharing information on your procedures. I agree that sufficient care has been provided with regards to preserving the participants' privacy.
> > - Q4: Thanks for the clarification.

---

### Official Review · Reviewer_wR6w · 2025-11-01

**Soundness:** 2
**Presentation:** 2
**Contribution:** 2
**Rating:** 4
**Confidence:** 3

**Summary:**

This paper introduces a new benchmark designed to move AI safety beyond binary harm classifications and toward a more pluralistic understanding. The authors present a scalable framework for generating prompts that systematically vary along two axes: harm severity (from benign to harmful) and human agreement (from consensus to disagreement). The benchmark contains 150 prompts, each with ratings from 100 human annotators, and is enriched with comprehensive annotator traits (demographic and psychological) and prompt-level features (actions, effects, values). Through extensive statistical analysis, the paper demonstrates that both prompt content-- especially tangible, imminent risks--and annotator traits jointly shape harm perception, with disagreement emerging from structured interactions between them. Finally, the authors benchmark several AI safety models and alignment methods, showing that personalized alignment significantly outperforms traditional consensus-based approaches, highlighting the need for systems that can adapt to diverse human viewpoints.

**Strengths:**

- The inclusion of deep annotator profiles, incorporating psychological measures like the Moral Foundations Questionnaire and Schwartz Value Survey, and structured prompt features from SafetyAnalyst and Kaleido provides a useful resource for analysis. This rich data enables the paper's core investigations into the drivers of disagreement, moving beyond simple demographics to the underlying values and psychological traits of annotators.
- The paper provides strong empirical evidence for a central hypothesis in pluralistic AI: that personalized alignment is superior to consensus-based alignment. The evaluation in Section 5 shows that for multiple models and methods, predicting individual ratings results in lower MAE than predicting aggregated ratings (Figure 8). This finding directly challenges the common practice of treating annotator disagreement as "statistical noise to be averaged out".

**Weaknesses:**

- The paper's core methodological claim is the creation of a "calibrated" harm spectrum by prompting an LLM (DeepSeek) to generate variants along a 0.0 to 1.0 ordinal scale. This process implicitly assumes the LLM is a neutral tool for varying a single latent dimension of "harm." However, the LLM itself is a complex model that may introduce systematic stylistic artifacts (e.g., changes in syntax, vocabulary, or tone) that correlate with the requested harm level. These artifacts could act as confounding variables. While the human validation shows a significant correlation (Spearman r = 0.59 in Figure 3), it does not rule out the possibility that humans are partially responding to these stylistic cues rather than solely the intended semantic variation in harm. The study could be improved by controlling for or analyzing these potential linguistic confounders to ensure the benchmark's harm axis is not entangled with unintentional stylistic variations introduced by the generator model.

- The analysis in RQ1 uses features extracted by SafetyAnalyst and Kaleido to predict human harm judgments. Finding that a feature like "Child Harm" extracted by SafetyAnalyst is a top predictor for human harm ratings (Figure 4) is somewhat circular. It demonstrates a strong correlation between one safety model's output and human labels, but it provides less independent explanatory power about the fundamental components of human judgment than implied. The analysis essentially confirms that a model built to identify specific harms generates features that align with human perceptions of those same harms. A more insightful approach might involve using more foundational, model-agnostic linguistic features to see if they can predict human ratings, thereby avoiding the conceptual loop of using one harm model to explain another (i.e., humans).

- The paper concludes that GPT-4.1 is more accurate than specialized safety models like WildGuard and SafetyAnalyst, attributing this to the "advantages of a more capable base model". This interpretation glosses over the vast difference in model scale. The comparison is between a massive, state-of-the-art frontier model and much smaller, open-source models. It is therefore unsurprising that the larger model performs better, especially in a few-shot, in-context learning setting. The experiment does not provide a fair comparison of a "general" vs. "specialized" model architecture or training objective, but rather a comparison of model scale. A more rigorous experimental design would involve fine-tuning the smaller specialized models on the personalized training data to compare their adaptability against GPT-4.1's in-context learning, which would provide a more nuanced understanding of the trade-offs between model scale, specialization, and alignment techniques.

- The study's design involves each annotator rating all 150 prompts in a single session, with a median completion time of 65 minutes. This block design is susceptible to ordering effects, where exposure to one prompt can prime or influence the judgment of subsequent prompts. The analysis models harm judgments as a function of stable annotator "traits" (demographics, psychological profiles) but does not account for transient psychological "states" (e.g., fatigue, desensitization, mood) that could evolve over the course of the long annotation task. Given the low variance explained by the annotator trait model (R^2 = 0.0231), these unmodeled, state-dependent effects could be a significant source of variance. Randomizing prompt order per participant and modeling for potential order effects would be necessary to more accurately isolate the influence of stable traits from the noise introduced by the experimental procedure itself.

**Questions:**

None

---

> ### Author Response · Authors · 2025-11-17
> **Response to Reviewer wR6w01**
>
> Thank you for the thoughtful and constructive feedback. We address each comment in turn.
>
> ---
>
> ## **Comment 1: Potential linguistic confounders**
>
> To evaluate whether DeepSeek introduced stylistic artifacts correlated with harm level, we performed an additional analysis using LIWC. We selected 15 stylistic categories and computed correlations with mean human harmfulness ratings. **None were statistically significant after Holm–Bonferroni correction.** The largest raw correlations were modest (e.g., *polite*: –0.223; *tentat*: 0.200; all corrected p > 0.07), indicating only weak, noisy trends. Given the short length of prompts and the coarseness of LIWC, these null results suggest that *systematic stylistic confounds do not substantially drive human ratings*. We will include this analysis in the camera-ready version.
>
> | liwc_category | pearson_corr | p     | p_holm |
> |---------------|--------------|-------|--------|
> | polite        | -0.223       | 0.005 | 0.077  |
> | tentat        | 0.200        | 0.012 | 0.170  |
> | tone_neg      | 0.172        | 0.032 | 0.416  |
> | cause         | 0.149        | 0.063 | 0.758  |
> | adverb        | 0.130        | 0.107 | 1.000  |
> | want          | 0.077        | 0.342 | 1.000  |
> | need          | 0.066        | 0.411 | 1.000  |
> | certitude     | 0.064        | 0.424 | 1.000  |
> | moral         | 0.064        | 0.427 | 1.000  |
> | allnone       | -0.051       | 0.525 | 1.000  |
> | negate        | -0.029       | 0.718 | 1.000  |
> | adj           | 0.029        | 0.721 | 1.000  |
> | curiosity     | -0.027       | 0.739 | 1.000  |
> | tone_pos      | -0.025       | 0.753 | 1.000  |
> | lack          | 0.019        | 0.816 | 1.000  |
>
> ---
>
> ## **Comment 2: Alleged circularity in using SafetyAnalyst/Kaleido features**
>
> We respectfully disagree that this analysis is circular. RQ1’s goal is *practical rather than psychological*: to decompose human harm judgments into **interpretable components that AI safety models can use**, not to recover the cognitive process behind human judgment. SafetyAnalyst and Kaleido provide structured, human-interpretable harm and value features that are precisely the kinds of dimensions we want AI systems to reason over. Thus, using them is appropriate rather than circular.
>
> Our results also offer the **first validation of these models on ambiguous, borderline safety cases**, where disagreement is highest and where existing safety tools have not been systematically evaluated. This demonstrates that their decompositions remain meaningful in the hardest region of the harm spectrum.
>
> To contextualize this further and rule out surface-form confounds, we added a PCA baseline using Qwen3-4B activations (Appendix D.5):
>
> - **first 42 PCs (matched dimensionality)**: R² = 0.339 (vs. 0.273 for our interpretable model).
> - **first-119-PC (95% variance) ceiling model (70 PCs after lasso)**: R² = 0.391.
>
> High-dimensional embeddings explain only modestly more variance than our interpretable features, while lacking explainability and generalizability. This reinforces the relevance of the SafetyAnalyst/Kaleido features.
>
> ---
>
> ## **Comment 3: GPT-4.1 vs. specialized safety models**
>
> We revised the manuscript to avoid implying architectural superiority and now frame the comparison as **general-purpose vs. specialized** models. We also expanded our k-shot evaluation to a broad range of models (GPT-5; Claude 3/3.5/3.7/4/4.5 variants; Qwen3 4B–32B). As shown in Table 1 of the updated manuscript:
>
> - Smaller models often failed to output even a short numeric prediction within 1024 tokens, highlighting practical limitations without specialization.
> - Newer Claude models (4.5 Sonnet, 4 Opus) sometimes refused harmless predictions due to safety filters.
> - GPT-5 exhibited the strongest performance overall.
>
> Regarding fine-tuning: personalized finetuning is infeasible with ~100 examples per annotator and too costly for real-world personalized alignment use. This is why our personalized alignment experiments focus on the more cost-effective method of *in-context* steering.
>
> ---
>
> ## **Comment 4: Ordering effects and annotator state**
>
> The following clarification and new analysis address this concern:
>
> 1. **Prompts were fully randomized per annotator**, eliminating systematic ordering (now clarified in Appendix D.1).
> 2. We added a **trial index random effect** to all models to capture potential mental state drifts due to fatigue or desensitization across the session.
> 3. **All RQ1–RQ4 results remained qualitatively unchanged** after including this random effect, indicating minimal impact of transient mental state.
>
> These updates are incorporated into the revised manuscript.
>
> ---
>
> Thank you again for the constructive feedback. We hope these clarifications and additional analyses address your concerns and strengthen the contribution, and we would be grateful if you would consider updating your assessment accordingly.

---

> > ### Comment · Reviewer_wR6w · 2025-11-26
> >
> > I thank the authors for their detailed response. The additional LIWC analysis and the PCA baseline effectively address my concerns regarding potential linguistic confounders and the validity of the feature analysis. I have increased my score.

---

### Official Review · Reviewer_EMxJ · 2025-11-04

**Soundness:** 2
**Presentation:** 2
**Contribution:** 3
**Rating:** 6
**Confidence:** 4

**Summary:**

This paper introduces the benchmarks Pluriharms to study how people with diverse backgrounds and values judge harmfulness, emphasizing disagreement aspects in safety evaluation.

The authors construct the dataset through a multi-stage process: generating over 60,000 prompts spanning a fine-grained harm spectrum using LLMs, extracting interpretable harm- and value-related features, and finally creating 150 diverse, disagreement-prone prompts for human annotation. Each prompt receives ratings from 100 participants whose demographic and psychological traits (values, morality, empathy, AI literacy, etc.) are also collected.

Their findings show that both prompt features (such as physical vs. psychological harm, rights, and duties) and annotator traits (e.g., education, political believes, exposure to online toxicity) shape harmfulness judgments.
When evaluating safety models  such as WildGuard and SafetyAnalyst on Pluriharms, they find that ''personalized alignment methods'' that are specific to individual annotators’ value profiles outperform general approaches ignoring annotators' traits, suggesting that pluralistic alignment better captures real human diversity.

Finally, they conclude that safety assessment should model pluralism and disagreement rather than averaging it out.

**Strengths:**

Dataset (creation)
* The process is well-explained. They achieve broad coverage and controlled variation across harm levels.
* Through annotation with specialized safety rating models, they add semantic structure, linking prompts to human-understandable ethical dimensions.
* Each participant rated all prompts. This allows within-subject comparisons of how individual traits affect harm judgments.
* The correlation between synthetic harm levels and human ratings supports the validity of the synthetic harm generation procedure.

Analysis
* Modeling trait × trait and trait × prompt reveals how disagreement stems from structured social and cognitive differences rather than noise.

Experiments
* Including both binary and probabilistic safety outputs (e.g. for WildGuard) provides insight into how uncertainty affects safety prediction.
* The profile-based alignment approaches are well motivated and improve model-human alignment.

**Weaknesses:**

Dataset (creation)
* The dataset is limited in size, covers only English prompts, and considers limited demographic diversity
* The focus on single prompts rather than (more realistic) full human–AI conversatinos/interactions.
* The reliability of the model-based feature extraction (SafetyAnalyst, KALEIDO) is assumed instead of validating themselves, potential biases in those models could affect prompt selection and impact the dataset curation process.
* The paper does not contain any details on data quality control for human ratings, such as attention checks or interrater reliability.
* Can you describe more clearly describe how the LLM was prompted to produce controlled harm variants?

Analysis
* The variance (~ 27% for prompt features, ~2–7% for traits and interactions) is noteworthy, but the discussion does not address what drives the remaining unexplained variance.
* The section presents many correlations and models but gives little conceptual synthesis connecting quantitative results to theories of moral judgment or harm perception.
* The analysis relies on automatically extracted features (from SafetyAnalyst and KALEIDO) which are not validated, so the interpretability of coefficients depends on potentially noisy input features.
* Sec 4 would benefit from qualitative examples showing how specific prompts or participant profiles led to divergent ratings.

Figures
* Readability of figures can be improved, e..g Fig 4: difficult to align x axis labels to their respective bars, Fig 5: captino is not mentioning what Factor 1-3 are but one needs to search in the text, Fig 6: the placement of the legend, etc.

Experiments
* The dataset used for evaluation is quite small & might  generalizability.
* It is unclear how the “value profile” steering process was conducted (e.g., whether GPT-4.1 generated summaries consistently or varied across runs) - more details would be useful.
* There is little discussion on wht personalized methods outperform aggregated ones — the analysis remains descriptive rather than diagnostic. What insights can we extract from these experiments and how can they guide development of future safety evaluators?

**Questions:**

See section weaknesses for questions

---

> ### Author Response · Authors · 2025-11-29
> **Response to Reviewer EMxJ (Part 1)**
>
> We thank the reviewer for the thoughtful and constructive feedback. Below, we address each point in turn.
>
> ---
>
> # **Dataset**
>
> ### **Weaknesses 1 & 2**
>
> We acknowledge that the dataset is limited in scale, linguistic coverage, and demographic diversity, and we explicitly note these limitations in the paper. While larger and more diverse datasets are always valuable, our design intentionally prioritized **per-annotator depth**: each participant rated **all 150 prompts**, yielding a rich within-subject dataset that enables individualized analyses rarely possible in safety benchmarks.
>
> Importantly, our results show that **substantial disagreement emerges even among annotators with similar demographics** (e.g., U.S.-based English speakers). Our framework provides a systematic method for *eliciting* these disagreements and opens the door to future work expanding across languages and cultures.
>
> Regarding the focus on single prompts rather than multi-turn conversations: collecting longer interactions would significantly reduce the number of examples each annotator could rate, directly trading off with our core goal of maximizing *per-annotator coverage*. Multi-turn settings are an important future direction, but the individualized analyses presented here would not be feasible with fewer per-person datapoints.
>
> ---
>
> ### **Weakness 3**
>
> We thank the reviewer for raising this concern and have added two new analyses to strengthen this point.
>
> 1. **External validation of SafetyAnalyst/KALEIDO features**
>    We conducted an experiment where each (prompt, feature) pair generated by SafetyAnalyst and KALEIDO was independently evaluated by GPT-4.1 (temperature 0). Agreement rates were:
>
>    - **ACTION:** 63.14% (466/738)
>    - **EFFECT:** 82.22% (370/450)
>    - **VALUE:** 77.70% (209/269)
>
>    While not perfect, a frontier model agrees with the *majority* of the specialized safety models’ predictions across all feature types, suggesting that these decompositions are reasonably robust. We acknowledge remaining noise and will explore improved feature robustness in future work.
>
> 2. **Data-driven evidence of feature utility**
>    As shown in **Appendix D.5** : our interpretable harm- and value-feature model explains a substantial portion of variance (R² = 0.273), performing competitively with high-dimensional PCA baselines (R² = 0.339 for matched dimensionality; 0.391 ceiling). This demonstrates that despite some noise, these features capture *strong and meaningful signal* relevant to human judgments, despite noise.
>
> ---
>
> ### **Weakness 4**
>
> Our study included explicit quality-control measures. As detailed in **Appendix D.1**, we implemented **three attention checks** and excluded participants who failed any of them. We also embedded **six gold-standard items** (three unambiguously harmful and three benign). Ratings showed clear separation:
>
> - **Harmful:** mean = 91.99 (95% CI [89.25, 94.74])
> - **Benign:** mean = 6.60 (95% CI [3.95, 9.26])
>
> We have added these statistics to the revised manuscript in Appendix D.1. Since the dataset intentionally emphasizes borderline cases to elicit disagreement, this gold-standard separation provides the strongest feasible measure of reliability without undermining our design.
>
> ---
>
> ### **Weakness 5**
>
> The full prompting scheme is included in **Appendix A**, including the system prompt, reasoning steps, and instructions for constructing Levels 0.0–1.0. In brief:
>
> 1. The LLM identifies the core topic of a reference prompt.
> 2. It defines a clearly benign variant (Level 0.0) and a clearly harmful variant (Level 1.0) using a fixed set of risk categories.
> 3. It generates intermediate variants that monotonically increase specificity toward potentially harmful applications while preserving the topic.
>
> Examples are provided in **Appendix B**.
>
> ---
>
> # **Analysis**
>
> ### **Weakness 1**
>
> We believe much of the remaining variance reflects **human rating noise**, which is unavoidable when using a **0–100 continuous scale**. People are not naturally calibrated to provide fine-grained numeric judgments.
>
> We include a baseline in **Appendix D.5** using **Qwen3-4B activation PCs**, which explain only **34–39%** of the variance depending on dimensionality. This provides a practical upper bound for any linear, interpretable model in this setting. Against this ceiling, our prompt-feature model’s ~27% represents strong performance, especially given that we intentionally curated **highly diverse, borderline prompts** where variance naturally concentrates at the prompt level.
>
> ---
>
> ### **Weakness 2**
>
> While our results relate to theories of moral judgment, testing moral psychology frameworks lies outside the scope of this dataset-focused paper. Our goal is **practical and methodological**: identifying *AI-compatible, interpretable features* for pluralistic safety modeling, not adjudicating between moral theories. We have clarified this motivation in the revised discussion.

---

> ### Author Response · Authors · 2025-11-29
> **Response to Reviewer EMxJ (Part 2)**
>
> ### **Weakness 3**
>
> See our response to Dataset - weakness 3.
>
> ---
>
> ### **Weakness 4**
>
> We already include qualitative examples in **Appendix D.3**, showing **high-variance prompts** alongside the **distribution of participant ratings**. These illustrate how specific prompts lead to divergent judgments.
>
> ---
>
> # **Figures**
>
> We have improved figure readability as suggested:
>
> 1. **Figure 4:** Added thin dashed vertical guides to align x-axis labels with bars.
> 2. **Figure 5:** Added concise descriptions of Factors 1–3 directly in the caption.
> 3. **Figure 6:** Moved the legend to the top-right corner for clarity.
>
> ---
>
> # **Experiments**
>
> ### **Weakness 1**
>
> While the evaluation set may appear small, **150 ratings per annotator** is substantial for personalized alignment. In realistic deployment, collecting even dozens of user-specific ratings is costly, so sample efficiency is key. Participants spent an average of **65 minutes** on the task—comparable to PRISM (≈70 minutes).
>
> ### **Weakness 2**
>
> As described in **Appendix E**, GPT-4.1 generates **one value profile per run**. The summaries are conditioned on the specific alignment examples for that annotator; thus, summaries differ across runs because the underlying examples differ.
>
> ### **Weakness 3**
>
> We have added a synthesis paragraph to **Section 5** summarizing the key mechanisms:
>
> 1. **Annotator traits and their interactions** contribute systematic variation in harmfulness judgments, which aggregated methods obscure.
> 2. **K-shot in-context learning** outperforms value-profile summaries, suggesting limitations in current summarization-based approaches and motivating development of more expressive preference-extraction techniques.
>
> ---
>
> We thank the reviewer again for the helpful feedback and believe the revisions significantly strengthen the clarity and rigor of the paper.

---

### Meta-Review · Area_Chair_U6MN · 2026-01-12

**Summary:**

Through 150 diverse disagreement prone prompts annotated by 100 participants, this paper shows that annotator traits (education, politics, exposure) shape harmfulness judgements and impact safety evaluations of models.  The main concerns raised are dataset size, reliability of model-based features, data quality control, and empirical insights e.g. model performance as a function of size. However, the paper is also acknowledged for several strengths: a new challenging benchmark that opens a new dimension for pluralistic safety evals; revealing how annotator disagreement stems from structured social and cognitive differences rather than noise,  that personalized alignment is superior to consensus-based alignment; writing clarity.

**Reviewer Concerns:**

Most concerns are adequately addressed.

**Reviewer Scores:**

The paper has only one negative reviewer who received a very comprehensive response. Their acknowledgement suggests that they too have flipped to a positive score:  "I thank the authors for their detailed response. The additional LIWC analysis and the PCA baseline effectively address my concerns regarding potential linguistic confounders and the validity of the feature analysis. I have increased my score."

---

### Decision · Program_Chairs · 2026-01-26

Accept (Poster)